# Tracing ancient solar cycles with tree rings and radiocarbon in the first millennium BCE

Nicolas Brehm[1,2] ✉, Charlotte L. Pearson [1], Marcus Christl [2], Alex Bayliss [3], Kurt Nicolussi [4], Thomas Pichler[4], David Brown[5] & Lukas Wacker [2] ✉

The Sun drives Earth's energy systems, influencing weather, ocean currents, and agricultural productivity. Understanding solar variability is critical, but direct observations are limited to 400 years of sunspot records. To extend this timeline, cosmic ray-produced radionuclides like $^{14}C$ in tree-rings provide invaluable insights. However, few records have the resolution or temporal span required to thoroughly investigate important short-term solar phenomena, such as the 11-year solar cycle, or $^{14}C$ production spikes most likely linked to solar energetic particle (SEP) events. Here we present a continuous, annually resolved atmospheric $^{14}C$ record from tree-rings spanning the first millennium BCE, confirming no new SEP's and clearly defining the 11-year solar cycle, with a mean period of 10.5 years, and amplitude of approximately 0.4‰ in $^{14}C$ concentration. This dataset offers unprecedented detail on solar behavior over long timescales, providing insights for climatic research and solar hazard mitigation, while also offering enhanced radiocarbon calibration and dating accuracy.

Cosmogenic radionuclides are continually produced in Earth's atmosphere by the impact of highly energetic cosmic rays[1,2]. The rate at which they are produced depends on several factors such as the flux and energy spectrum of galactic cosmic rays, the level of solar activity, and the strength of the geomagnetic dipole field[1,3,4]. Changes in the intensity of galactic cosmic rays on Earth are strongly coupled with the 11-year solar cycle as observed in sunspots caused by the Sun's magnetic field reversal about every 22 years.

Solar activity from cosmogenic radionuclides can be quantified using the solar modulation parameter (Φ), which reflects the degree of solar magnetic shielding affecting galactic cosmic rays[1,2,5]. Radionuclides, such as radiocarbon ($^{14}C$) found in tree-rings or $^{10}Be$ preserved in ice cores, serve as reliable indicators for reconstructing solar activity. While several annually resolved $^{14}C$ records have emerged recently, they often fall short in length[6–15], hindering systematic studies of short-term solar variability such as 11-year and 22-year cycles in the past. Recently, annual $^{14}C$ tree ring data were obtained over the last millennium[16] however, no millennia-long annually resolved records exist beyond the first millennium CE.

Here we introduce in detail the atmospheric concentration of $^{14}C$ over the first millennium BCE with annual resolution, based on dendrochronologically dated tree ring samples. When combing the $^{14}C$ data with the existing high precision record of Fahrni et al.[17], the dataset covers the time span from 1000 BCE to 2 BCE at annual resolution (Fig. 1). This record establishes a standard for precise and temporally accurate reconstructions of $^{14}C$ production and the solar modulation parameter over the studied time period and it allows for a systematic exploration of short-term solar variability, particularly the 11-year solar cycle. Moreover, it allows us to systematically test for the possibility of $^{14}C$ production spikes (which have recently garnered attention due to their association with solar energetic particle (SEP) events[18–23]) within this period. SEP events may pose significant risks to modern technological systems, both on Earth and in space[24]. This annual data marks a substantial advance by providing a significantly

[1]Laboratory of Tree Ring Research, University of Arizona, Bryant Bannister Tree-Ring Building, 1215 E. Lowell Street, Tucson, AZ 85721-0045, USA. [2]Laboratory of Ion Beam Physics, ETH Zurich, Otto-Stern Weg 5 HPK, 8093 Zurich, Switzerland. [3]Historic England, Cannon Bridge House, 25 Dowgate Hill, London EC4R 2YA, UK. [4]Department of Geography, Universität Innsbruck, Innrain 52, 6020 Innsbruck, Austria. [5]School of Natural and Built Environment, The Queen's University, Belfast BT7 1NN, UK. ✉e-mail: nbrehm@arizona.edu; wacker@phys.ethz.ch

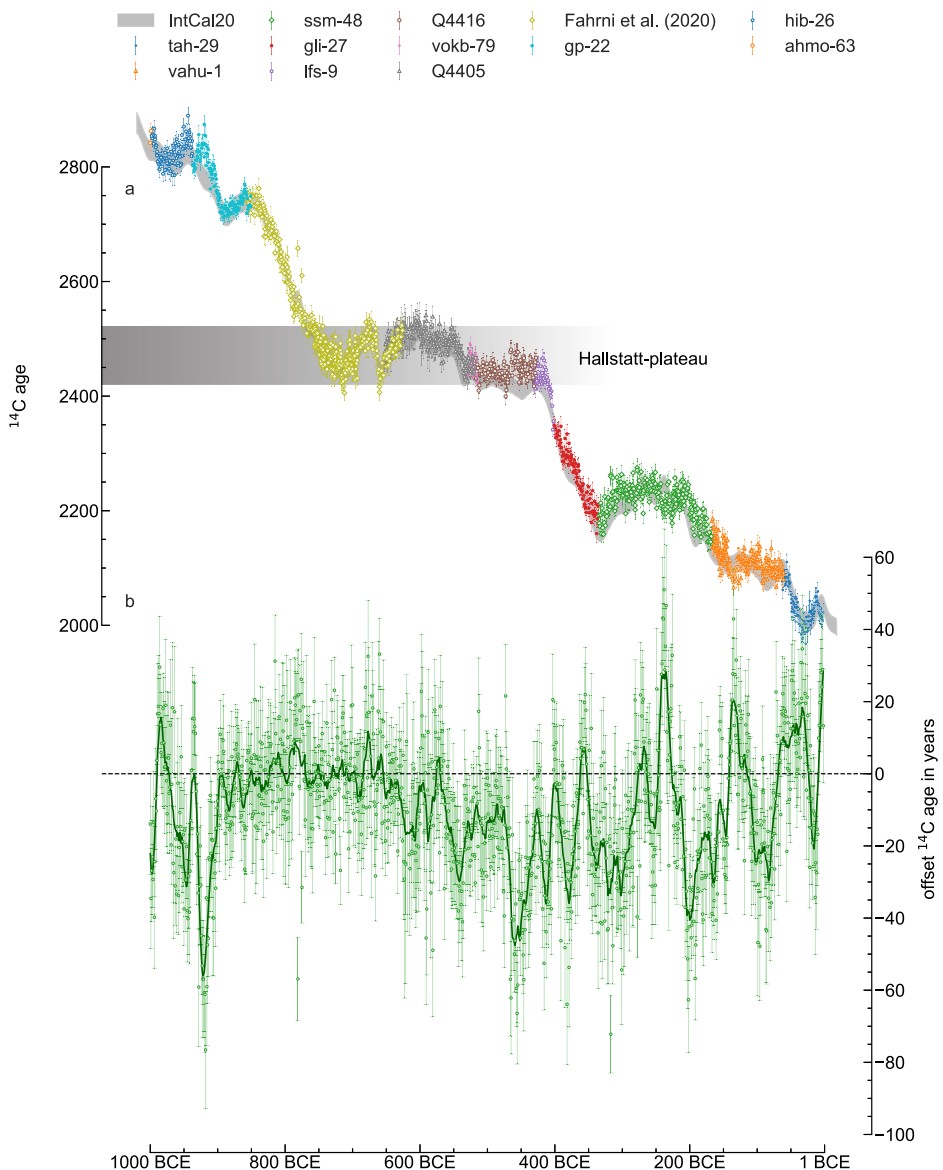

**Fig. 1 | Overview of the measured ¹⁴C data from tree rings. a** Individual measurement results in radiocarbon age before the year 1950 of each tree with 2-σ error bars. **b** Difference between the annual data and the IntCal20 calibration curve in ¹⁴C age. Error bars indicate the 2-σ error bars. The green line shows a Savitsky-Golay smoothed curve.

longer temporal span than has previously been available in single year ¹⁴C across this period. These data can reveal insights into solar cycles and can also be used to significantly enhance the potential for effective use of radiocarbon dating in this time period, particularly between ca. 750 and 420 BCE, by adding sub-decadal structure to data available for radiocarbon calibration. This period, known as the Hallstatt plateau or 'the 1st millennium BC radiocarbon disaster' has previously posed a significant hurdle to the construction of accurate radiocarbon dated chronologies for the study of archaeology and environment at sites as far apart as China[25], Southeast Asia[26], South America[27], Germany[28] the Mediterranean[29] and the Balkans[30]. Any single radiocarbon determination from between ca. 750 and 420 BCE measures as c. 2520–2420 radiocarbon years BP, reflecting the plateau (Fig. 1). Attempting to calibrate such dates using the IntCal20 calibration curve[31] frequently gives inconclusive results, with calibrated time ranges spanning the entire period. This situation can be somewhat mediated by the use of Bayesian models[32,33] but also by reducing the levels of uncertainty within the calibration dataset crossing the plateau[17]. Currently the

structure of the majority of the calibration curve for this time period is made up of measurements on multi-year blocks of tree-rings, averaging/dampening any annual signal. By providing single year atmospheric ¹⁴C data across the period we can more accurately describe past variations, revealing sub-decadal structures which may provide a more accurate fit for Bayesian models and opportunities to pattern-match other single year data from tree-rings dating to this temporal range. As such, this data offers significant potential to significantly increase the accuracy and precision of radiocarbon wiggle matching and other Bayesian chronological models that include informative prior information[34].

## Results

### Annual ¹⁴C record over the first millennium BCE

State-of-the-art compact accelerator mass spectrometry was applied to an absolutely dated tree ring archive producing a total of 1127 high-precision ¹⁴C measurements on 824 individual dendrochronologically dated tree-rings from Ireland, and the Alps (Supplementary Section 1,

Supplementary Fig. 3). The data cover the two time periods from 1000 BCE to 851 BCE and 652 BCE to 2 BCE at annual resolution and were combined with a previously published single year record covering the period 856 to 626 BCE[17] to produce a 1000-yr-long sequence (1000 BCE – 2 BCE) (Fig. 1). The new data overall agrees well with the previously produced, lower resolution, IntCal20 curve (Fig. 1a), however the additional detail provided by the single year data highlight some notable differences around the years 900 BCE and 425 BCE, where an absolute deviation of more than 25 [14]C years is observed (Fig. 1b). Otherwise, the additional structure of the data offers several ways to tackle the radiocarbon dating problem with the Hallstatt plateau. This data therefore could have a significant impact on the dating accuracy possible for a wide range of archaeological and environmental studies.

### Reconstruction of solar activity

The high-precision dataset not only refines our understanding of past variations in radiocarbon and so has potential impact for dating in this period but also provides a crucial foundation for reconstructing past solar activity. Global [14]C production was derived from the $\Delta^{14}$C data (Fig. 2a) using a carbon cycle 22-box model[16] to estimate historical solar activity (Supplementary Fig. 4). To mitigate the effect of the [14]C production event in the year 664 BCE[19] on the solar reconstruction, the event was simulated with the carbon cycle box model by fitting a production spike in the year of interest. The same simulation was then run without the production spike. Afterwards the $\Delta^{14}$C data was detrended by the difference of the two simulations (Supplementary Fig. 2). No additional error was propagated to the detrended data. This correction was applied because the [14]C production of the 664 BCE event is probably not caused by galactic cosmic rays, but instead by solar energetic particles. Production of [14]C varies by ±30% over the whole period. The solar modulation parameter $\Phi$ was calculated from the [14]C production derived from the $\Delta^{14}$C record by using the conversion table computed by Herbst et al.[2] (Supplementary Fig. 5) and by using the geomagnetic field record of Panovska et al.[35].

For comparison with other lower resolution records of solar modulation parameters[36,37], we used a 50 year lowpass filter for our reconstruction to match the resolution of these records. This reconstruction of solar modulation parameter correlates well with previously existing reconstructions which are based on lower resolution data[36] ($r = 0.85$, $p = 9.4e-14$)[37] ($r = 0.86$, $p = 3.5e-30$) (Fig. 2c). The studied time span covers two grand solar minima which are clearly visible in the 50-yr low pass filtered record (blue shaded area in Fig. 2). The two solar minima show a prolonged reduction in solar activity by 55% and 65% during the two time periods 833 BCE to 705 BCE and 413 to 325 BCE compared to the mean $\Phi = 590 \pm 75$ MeV of the rest of the studied period. We can also observe an apparent reduction in solar modulation around the 664 BCE event in both the Wu and Steinhilber records, because neither of these consider the 664 BCE event as a solar event.

### Solar cycle detection in high-resolution radiocarbon records

The obtained high resolution record of solar modulation also carries the high frequency signal of the 11-year solar cycle. To confirm this signal in the annual radiocarbon record, an in-depth spectral analysis was performed on the detrended $\Delta^{14}$C record (Fig. 3a). To limit the analysis to the short periods of interest, such as the 11-year and 22-year cycles, the $\Delta^{14}$C data was detrended by subtracting the 30-year lowpass filtered $\Delta^{14}$C record. To get information about the amplitude and frequency of the 11-year solar cycle through time a Morlet wavelet transform was calculated, where the areas of more than 95% significance were marked. The wavelet analysis shows that a statistically significant signal of around 11 years as well as 22 years can be detected during nearly the whole time span (Fig. 3b). A simple fast Fourier transform on the detrended data also reveals a significant (> 99.9%

false alarm level) mean period of about 10.5 years with an amplitude of about 0.4‰ (Fig. 3c).

Additionally, a Lomb–Scargle periodogram[38] was computed for different time-ranges of the $\Delta^{14}$C data (Fig. 3). The 99.9% false alarm threshold was calculated for the Lomb–Scargle spectra by Monte-Carlo simulations. We observed a main significant peak at the period of 10.6 years accompanied by a smaller peak at 10.3 years when analyzing the full record. We also observed several significant peaks around 22 years, corresponding to the Hale cycle.

It has been suggested via observations of sunspot data that the 11-year solar cycle is significantly reduced during solar minima. More recently, this has been systematically tested through single year [14]C data and the same effect was observed[16]. To investigate this further, a Lomb–Scargle analysis, well known for the detection and characterization of periodic signals in unevenly sampled data[38], was applied solely on the data from 833 BCE to 705 BCE and 413 to 325 BCE, covering the two grand solar minima within the study period. All the significant periods between 5 and 30 years vanished when only analyzing data from these two minima (Fig. 3e). This effect could not be reproduced when selecting two random time intervals of the same length as the two solar minima, which confirmed that the disappearance of the cycles in the minima was not an artifact of shortening the data series. This finding was also visible in the wavelet analysis, where, again, during the grand solar minima we observed that the 11-year solar cycles seem to vanish (Fig. 3b). On the other hand, when the Lomb–Scargle analysis was applied to the record excluding the two grand solar minima, the significant peaks of the whole time period were enhanced, with significant peaks appearing at 11.1 and 13.9, revealing more details about fluctuations in the short term 11-year cycle (Fig. 3f). While these observations seem compelling, we note that Lomb–Scargle analysis has some limitations for the evaluation of quasi-periodic signals such as the 11–year cycle. We also note that while the appearance of 22-year cycles in the wavelet analysis seems to potentially contradict an argument for the absence of 11-year cycles, instead it may indicate that one of the solar magnetic polarities simply has a significantly stronger shielding effect from the galactic cosmic rays than the other.

### [14]C production events

The annual [14]C record was also systematically screened for previously undetected rapid SEP-like increases. The $\Delta^{14}$C was detrended in the same way as for the spectral analysis, and the difference between the 3-year average before and after every year was calculated (Fig. 4). A 3-σ threshold of 4.5‰ was chosen to search for new potential [14]C events. The only significant increase was detected in the year 663 BCE, which corresponds to the event already found in 664 BCE. It is notable that no further significant [14]C production spikes besides the known 664 BCE event were found within the period, meaning that either no additional extreme events occurred, or if they did, they were below the 3-σ threshold.

## Discussion

Our annual data was used to reconstruct past solar activity at an annual resolution from 1000 BCE to 2 BCE, giving further insight into solar dynamics far beyond the direct observational record. Overall a mean period of about 10.5 years with an amplitude of about 0.4‰ was found. This corresponds to a relative production variation of about 4% when using the carbon cycle box model. A more detailed spectral analysis shows an enhanced presence of the 11-year solar cycle during times of high solar activity. No significant 11-year signal was found during the two grand solar minima studied. This agrees well with indications from [14]C records covering the most recent millennium, where a significantly reduced solar cycle was found during grand solar minima[16]. Systematic testing of the data in this way confirms that the 11-year cycle is indeed

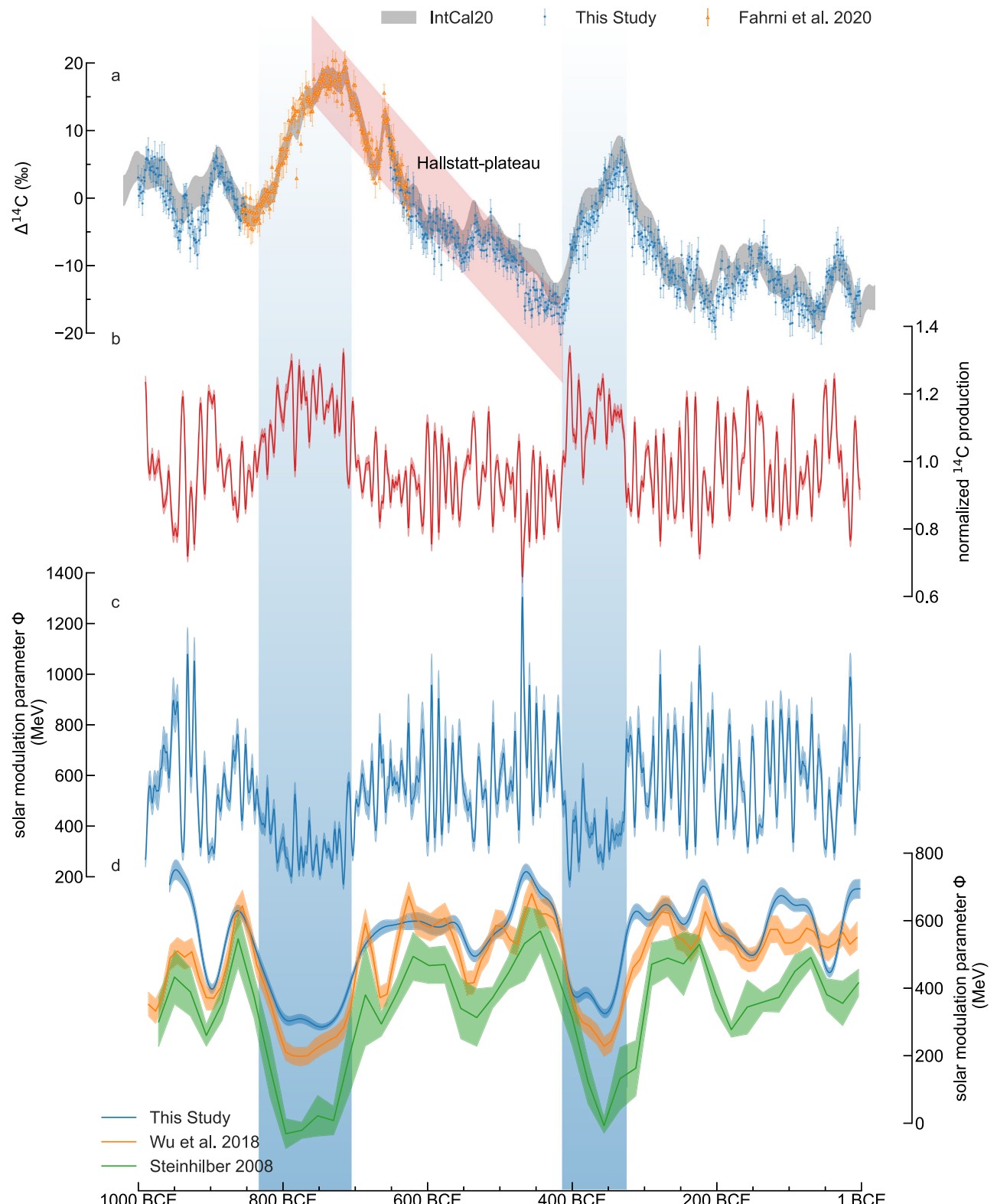

**Fig. 2 | From annual Δ¹⁴C records to solar modulation. a** Annual Δ¹⁴C records with 2-σ error bars. Δ ¹⁴C (‰) denotes the decay corrected ¹⁴C/¹²C ratio of a sample relative to a standard and normalized for isotope fractionation. **b** Normalized ¹⁴C production rates relative to the steady state production rate of 6.6 kg/year calculated with a carbon cycle box model. **c** Reconstructed solar modulation parameter Φ from ¹⁴C production rates. The shaded bands indicate the uncertainty estimated by Monte Carlo simulations. **d** 50-yr low-pass-filtered Φ compared to existing reconstructions of solar modulation[36,37]. The two grand solar minima are indicated by the blue shaded areas and the Hallstatt-plateau is indicated by the red shaded area.

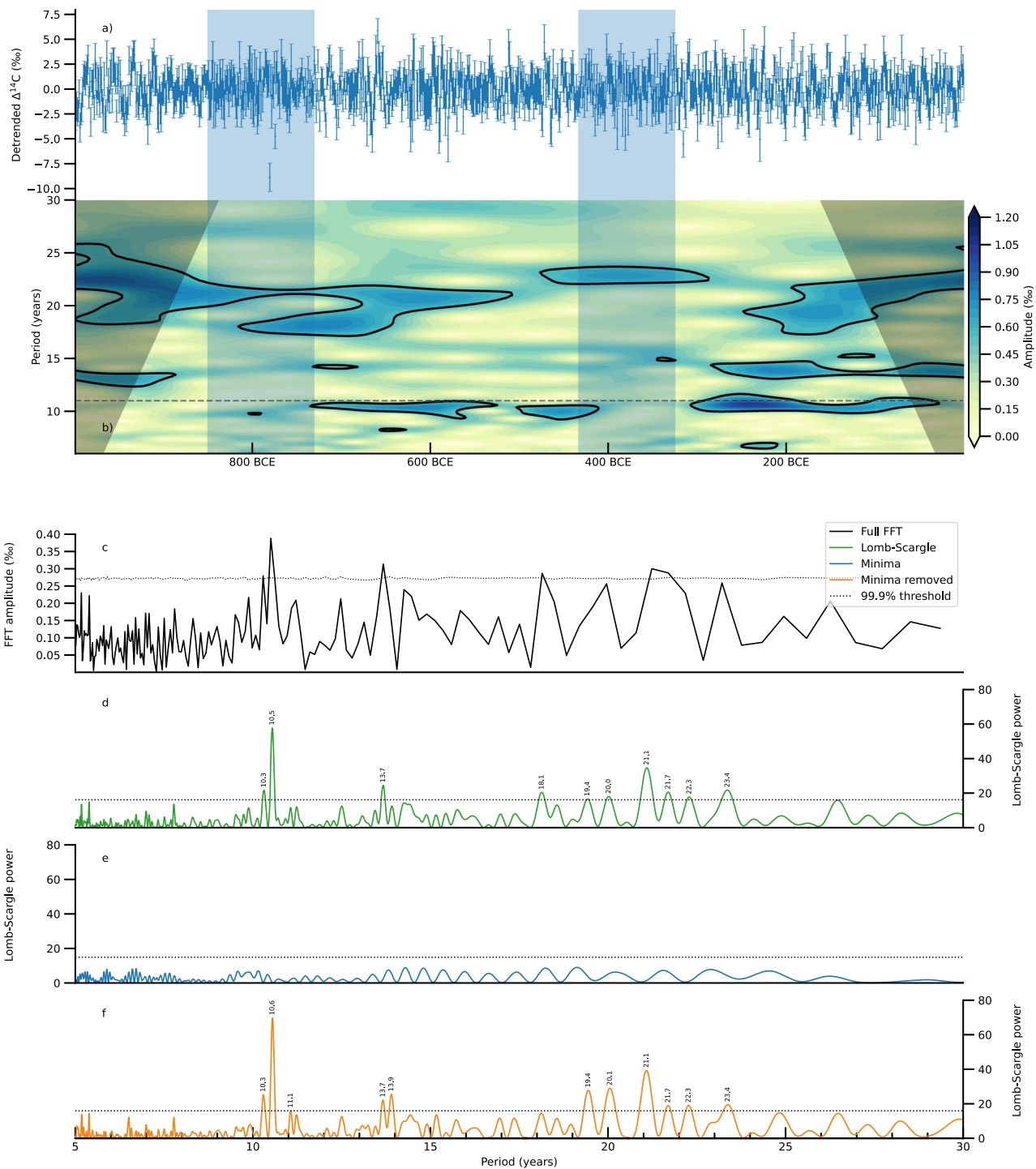

**Fig. 3 | Spectral analysis of the $\Delta^{14}$C record. a** 30-yr high pass detrended $\Delta^{14}$C record. The error bars indicate the 2-σ uncertainties. **b** Morlet Wavelet analysis of the 30-yr detrended $\Delta^{14}$C record. The black lines show significant regions above a 95% confidence interval. An 11-year period is indicated with the black striped line. **c** Fast Fourier Transform with the 95% confidence threshold of the record. **d** Lomb–Scargle periodogram of the full record. **e** Lomb–Scargle periodogram of the time periods of the two grand solar minima. **f** Lomb–Scargle periodogram of the record the two grand solar minima removed.

reduced or nonexistent during grand solar minima and provides new insights to assist in the forecasting of future solar cycles.

In addition to the 11-year cycle, several significant periods around 22-years were detected. This interesting finding indicates that one of the solar magnetic polarities has a significantly stronger shielding effect from the galactic cosmic rays than the other. If it was not the case, the 22-year cycle would not be visible in the $^{14}$C data. This confirms the model, that the relative flux for positive and negative

polarities depends on the level of solar activity which leads to the change in the variability of tilt angle[39]. During periods of low solar activity, such as the grand solar minima, the flux can be even higher at negative polarities[40]. This could explain why we can still detect a 22-year cycle during the solar minima when no significant 11-year cycles are detected. The 22-year cycle can also be observed in direct measurements of galactic cosmic rays today, where differences in galactic cosmic ray flux are observed, depending on the polarity of the solar

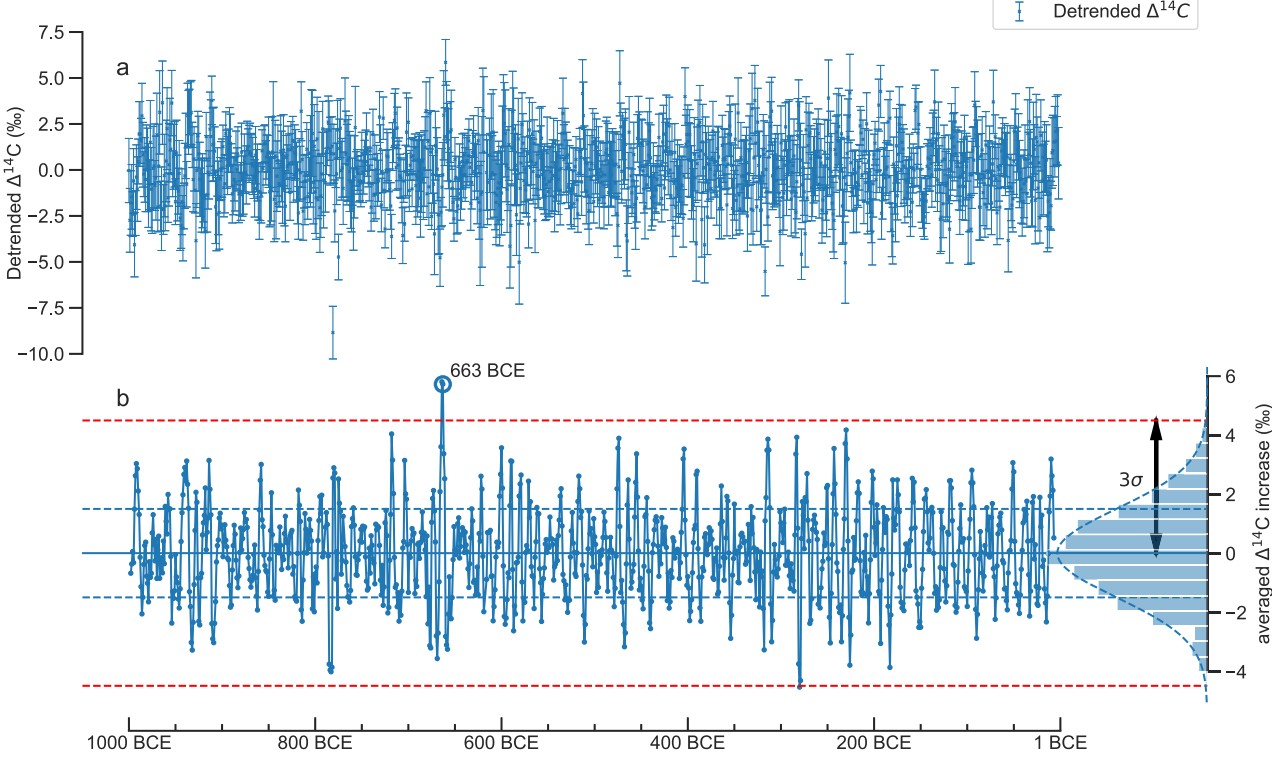

**Fig. 4 | Illustration of the search for SEP events. a** Detrended $\Delta^{14}C$ record (blue) with 2-σ uncertainties. **b** Changes between the 3-yr average before and after the event. The 3-σ range of about 4.5 ‰ is particularly emphasized for positive values, since this threshold value is exceeded only in 663 BCE during the first millennium BCE for two consecutive years.

magnetic field[41]. This further confirms the reliability of the annually resolved $^{14}C$ as solar proxy.

The presence of consistent 11-year cycles in the $^{14}C$ data during times of high solar activity has an entirely separate 'dual function' in that these data also present major new opportunities to fit short, single year sequences from tree-rings across radiocarbon plateaus. This could significantly improve the precision of dating provided by Bayesian chronological modeling of archaeological or environmental studies.

The data was also screened for rapid event-like increases. Although no new significant $^{14}C$ production spikes besides the already discovered 664 BCE event could be detected. This finding does still further improve our understanding about the frequency of occurrence and magnitude of such extreme $^{14}C$ events.

Future annual records of $^{14}C$ measured in tree rings will extend beyond this period, providing improved statistical constraints on solar dynamo models, solar cycle length and amplitude, and their relation to climate, as well as potentially revealing new smaller scale $^{14}C$ production events. Furthermore, additional records might also improve the signal-to-noise-ratio during grand solar minima to give further insights about the 11-year solar cycles during prolonged times of low solar activity. Longer single year tree-ring based $^{14}C$ will also provide new opportunities to act as a baseline for proxy development in combination with modern instrumental records.

## Methods
### Sample preparation and quality control
Our study focuses on the reconstruction of the past $^{14}C/^{12}C$ levels in the atmosphere from accurately dated tree ring chronologies[42]. Dendrochronologically dated trees represent the most reliable archive for the reconstruction of past atmospheric $^{14}C/^{12}C$ ratios and are therefore considered to be most suitable for a temporally accurate reconstruction of solar activity at the highest possible (that is, annual) resolution.

Dendrochronologically dated wood samples from Ireland, supplied by Historic England and from the Alps, supplied by the University of Innsbruck, were dissected into annually resolved samples weighing 30–60 mg (Supplementary Section 1). Typically, 54 tree ring samples with 4 wood blanks (2 Brown Coal (BCB) and 2 Kauri Stage 7 (K7B)) and two reference samples[43] each weighing 30–60 mg, were prepared in 15 ml glass test tubes together in a batch (making 60 in total). In a slightly modified procedure following Němec et al.[44], samples were first soaked in 5 ml 1 M NaOH overnight at 70 °C in an oven. Then the samples were treated with 1 M HCl and 1 M NaOH for 1 h each at 70 °C in a heat block, before being bleached at a pH of 2–3 with 0.35 M $NaClO_4$ at 70 °C for 2 h. The remaining white holo-cellulose was then freeze-dried overnight.

About 2.5 mg dried holo-cellulose was wrapped in cleaned Al capsules[45] and converted to graphite using the automated graphitization line AGE[46] A sample batch for accelerator mass spectrometry measurements consisting of 39 samples each was made up of three oxalic acid 1 (OX1) and four oxalic acid 2 (OX2) standards, 26 samples, two cellulose blanks, two chemical blanks and two reference samples (1515 CE, and 492 BCE). Individual cellulose preparations of the references were used for at least two measurement sets. Two accelerator mass spectrometry sample batches were typically prepared within a week and subsequently measured. Data analysis was performed with the ETH Zurich in-house data reduction software BATS[47]. Typically, a 1‰ external error for sampling and sample preparation was added to the finally reported uncertainties. The additional 1‰ was estimated by the long-term evaluation of the references used for quality control. For quality control the measurement of about every third sample was replicated on the same cellulose. This allows for a chi-square analysis and shows that the repeated measurements are generally in good agreement with one another. The resulting chi-squared values of the different chronologies are shown in Table 1. An overall chi-square

**Table 1 | Statistical analysis of the repeatability of the measurement of each individual tree**

| Tree | Total measurements | $\chi^2$ | Number of repeated rings | p-value |
|------|------|------|------|------|
| tah-29 | 81 | 16.2 | 19 | 0.65 |
| vahu-1 | 170 | 70.2 | 63 | 0.25 |
| ssm-48 | 242 | 58.7 | 74 | 0.9 |
| gli-27 | 88 | 25.5 | 22 | 0.27 |
| lfs-9 | 37 | 6.1 | 4 | 0.19 |
| Q4416 | 113 | 10.5 | 22 | 0.98 |
| vokb-79 | 21 | 4.7 | 6 | 0.58 |
| Q4405 | 174 | 29.3 | 42 | 0.93 |
| gp-22 | 117 | 22.2 | 30 | 0.85 |
| hib-26 | 82 | 30.1 | 21 | 0.09 |
| ahmo-63 | 2 | 0.0 | 0 | - |

Of the 11 obtained p-values, none of them are below the 95% confidence interval. For the ahmo-63 no repeated measurements were done and no p-value could be calculated.

analysis gives a chi square value of 302 with 329 degrees of freedom, giving a p-value of 0.85.

Two internal wood reference materials from CE 1515 and 492 BCE (pine and oak) and two different radiocarbon free wood blanks (K7B and BCB from Reichwalde) were repeatedly analysed together with the annual wood samples. Whereas the wood blanks were used for blank subtraction in the data evaluation process, the CE 1515 and 492 BCE references were used for quality control only.

Processing blanks samples typically yielded radiocarbon ages between 48 and 52 kyr without any background correction (see also Sookdeo et al.[43]). The results given for the 1515 CE reference are very consistent (Supplementary Fig. 3) with a mean radiocarbon age of 337 ± 2 yr BP (F[14]C: 0.9589 ± 0.0002), where the uncertainty of the mean was estimated by calculating the error of the mean. The sample scatter of 13 yr is slightly less than the mean uncertainty of 14 yr given by the applied data reduction ($\chi^2 = 61.0$, $n = 69$, $p = 0.74$). For the 492 BCE reference a mean radiocarbon age of 2440 ± 2 yr BP (F[14]C: 0.7381 ± 0.0002) was obtained. The sample scatter of 13 yr is also slightly less than the mean uncertainty of 16 yr given by the applied data reduction ($\chi^2 = 44.6$, $n = 52$, $p = 0.76$).

**Carbon cycle box model**

One of the main tools for the analysis of the measured $\Delta^{14}C$ records is the global carbon box model, which describes the transport of the $^{14}C$ produced in earth's atmosphere into other carbon pools such as the ocean and biosphere. The model is used to reconstruct the global atmospheric $^{14}C$ production rate from measured $\Delta^{14}C$ data and or simulate the resulting $\Delta^{14}C$ for specific $^{14}C$ production spikes, possibly from SEP events. For this, we use a carbon cycle box model based on the model of Güttler et al.[48] which was also used in Brehm et al.[16].

As the cycles of the two hemispheres are separated, the different mass distribution of biota and ocean on the hemispheres has to be considered. The northern hemisphere is covered by 70% of the global landmass, the southern hemisphere only contains 30%. The oceanic content of the northern hemisphere is 40% and of the southern hemisphere 60%. The stratosphere, the troposphere and the sedimentary sink were cut into halves. The initial $^{12}C$ masses of the global reservoirs taken from Güttler[48] were distributed accordingly (Supplementary Fig. 4).

Radiocarbon is produced in the stratosphere and the troposphere of both hemispheres, where 70% is produced in the Stratosphere and 30% in the Troposphere[48]. The fluxes between the biosphere and ocean to the troposphere were slightly adjusted to ensure a correct $\Delta^{14}C$

offset between the northern and southern troposphere. The model has a monthly time resolution, although seasonal variability of fluxes was not considered in the model.

The $^{12}C$ and $^{14}C$ content of each box after a time step is calculated with

$$N_i^{12}(t+\Delta t) = N_i^{12}(t) + \frac{dN_i^{12}}{dt}\Delta t \tag{1}$$

$$N_i^{14}(t+\Delta t) = N_i^{14}(t) + \frac{dN_i^{14}}{dt}\Delta t \tag{2}$$

$$\frac{dN_i^{12}}{dt} = \sum_j F_{ji}^{12} - \sum_j F_{ij}^{12} \tag{3}$$

$$\frac{dN_i^{14}}{dt} = -\lambda N_i + \sum_j F_{ji}^{14}(t) - \sum_j F_{ij}^{14}(t) + P_i(t) + P_{st,i} \tag{4}$$

Here $N_i^{12}$ and $N_i^{14}$ are the $^{12}C$ and $^{14}C$ content of the i-th box respectively, and $\lambda = 1/8267 yr^{-1}$ is the decay constant of $^{14}C$. The time step $\Delta t$ was chosen to be one month for all the following simulations. $F_{ji}^{14}$ and $F_{ji}^{12}$ describe the fluxes the $^{14}C$ and $^{12}C$ fluxes from box i to box j and are given by the following:

$$F_{ij}^{14}(t) = F_{ij}^{12}\frac{N_i^{14}(t)m_{14}}{N_i^{12}m_{12}} \tag{5}$$

Here $F_{ij}^{12}$ are the fluxes given in figure and $m_{12}$ ($m_{14}$) are the masses of $^{12}C$ ($^{14}C$) in atomic units. The $^{14}C$ production of each box is given by:

$$P_i(t) = V_i p(t), P_{st,i} = V_i p_{st}$$

$$V_i = \begin{cases} 0.5 \cdot 0.7, & if\, i = 0, 11 (Stratosphere) \\ 0.5 \cdot 0.3, & if\, i = 0, 11 (Troposphere) \\ 0, & else \end{cases} \tag{6}$$

Where $V_i$ distributes the total $^{14}C$ production p(t) or $p_{st}$ into the stratosphere and troposphere of the northern and southern hemisphere, where 70% is produced in the stratosphere and 30% in the troposphere. The steady state of the model was computed by simulating 200,000 years with the constant production rate pst = 6.6 kg/yr ($\Phi$ = 560 MeV, Earth dipole moment = 7.8·10$^{22}$Am$^2$). The model does not consider isotopic fractionation and thus the $^{14}C$ fluxes scale just as the $^{12}C$ fluxes. The $\Delta^{14}C$ of each box for the simulation is calculated by the following expression:

$$\Delta^{14}C_i(t) = \frac{\frac{N_i^{14}(t)}{N_i^{12}(t)} - k}{k} \cdot 1000, \tag{7}$$

where $k = \frac{N_{TN,st}^{14}}{N_{TN,st}^{12}}$ is the steady state ratio of $^{14}C$ to $^{12}C$ content of the troposphere of the northern hemisphere.

By evaluating the Eq. (7) at time $t+\Delta t$ and combining this with eqs. (1) and (2) an expression for the production can be extracted for all times for a given $\Delta^{14}C$ data record.

$$p(t) = \frac{\left(\frac{\Delta^{14}C_{Data,i}(t+\Delta t)k}{1000} + k\right)N_i^{12}(t+\Delta t) - N_i^{14}(t) - A_i(t)\Delta t}{V_i \Delta t}, \tag{8}$$

where

$$A_i(t) = -\lambda N_i^{14} + P_{st,i} + \sum_j F_{ji}^{14}(t) - \sum_j F_{ij}^{14}(t), \qquad (9)$$

and

$$N_i^{12}(t+\Delta t) = N_i^{12}(t) + \left(\sum_j F_{ji}^{12}(t) - \sum_j F_{ij}^{12}(t)\right)\Delta t \qquad (10)$$

The $\Delta^{14}$C data record must be interpolated to the same resolution as the simulation time step. For this the data was linearly interpolated with a monthly sampling. Since the differential equation is only solved in first order, the production can only be reconstructed for $\Delta^{14}$C records of boxes directly affected by the cosmic ray production, namely the stratosphere or the troposphere. Else $V_i$ in the denominator would be zero. Since our dataset covers atmospheric data recorded by tree-rings this is not an issue. For cases where the reconstructed production rate is negative, the production rate is set to zero because a negative production rate would physically not be possible. With this model, the $^{14}$C production rate can be obtained by calculating the total $^{14}$C production needed to reach the next measured $\Delta^{14}$C level at each time step. To get a reasonable start state at a given time, the production rate for the whole IntCal20 record has been calculated and the simulated state can be loaded at any time before 1950 CE.

To estimate the uncertainties, a set of 1000 Monte Carlo realizations of the data within the uncertainties of the measurements were generated. For each statistical realization of the data, a Savitzky-Golay filter[49] was applied and the production was reconstructed for the filtered data. The uncertainty of the production rate at every time step is then estimated by the standard deviation of all simulations.

### Reconstruction of solar modulation

The production of radionuclides depends on solar activity and the geomagnetic dipole field in a nonlinear way. To characterize the solar magnetic field we use the so-called solar modulation parameter $\Phi$, which is used to describe the shielding of the cosmic-ray flux by solar magnetic activity via the force-field approximation[2,5,50,51]. High values of $\Phi$ indicate a strong solar magnetic shielding, implying a strong deflection of galactic cosmic rays, thus yielding low production rates of cosmogenic nuclides. The reconstruction of solar activity (solar magnetic field) requires considering variations of the earth magnetic field. For the calculation of the solar modulation parameter a geomagnetic field record reconstruction by Panovska is used[35]. The magnetic field strength can be expressed in the form of the virtual axial dipole moment (VADM). For a given VADM and $^{14}$C production rate, the solar modulation parameter $\Phi$ can be estimated by considering the relation of production rate, solar modulation parameter and VADM that was computed by Herbst et al.[2] (Supplementary Fig. 5). The resulting table was calculated by performing a full Monte-Carlo simulation of the nucleonic component of the cosmic ray induced atmospheric cascade[2]. Consequently, the calculation of the solar modulation parameter assumes that all $^{14}$C is produced by galactic cosmic rays, where the intensity of galactic cosmic rays outside the heliosphere is assumed to be constant.

### Modelling and detrending of SEP events

To evaluate the additionally produced $^{14}$C caused by a potential SEP event observed, a gaussian shaped production spike with a full width half maximum of two months was introduced to an otherwise stable production rate. With this the global $^{14}$C production rate at time t can be described with:

$$p(t) = A e^{-\frac{1}{2}\frac{(t-t_0)^2}{\sigma^2}} + B \qquad (11)$$

The amplitude A and offset B of the of the production spike were adjusted such that the simulated $\Delta^{14}$C best fitted the data. The temporal position $t_0$ of the Gaussian production spike was fixed in the year of the event. Since the energy spectrum of the SEPs is softer than the one of the galactic cosmic rays the excess $^{14}$C produced by the event introduced by the Gaussian spike was mainly put into the stratosphere (90%) and only 10% into the troposphere. The choice of 90% stratospheric and 10% tropospheric production were made by rounding the calculated values from $^{14}$C production simulations (12.05% Troposphere and 87.95%)[52] with the assumption of a "hard spectrum" as proposed by O'Hare et al.[19].

## Data availability
The $^{14}$C data generated in this study are provided in the Supplementary Information. The Solar modulation data generated in this study are provided in the Source Data file. Source data are provided with this paper.

## Code availability
Any computer code used for evaluation of the results of this study will be available on request.

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

## Acknowledgements

N.B. is partly funded by the Swiss National Science Foundation (SNSF grant #SNF 197137). C.L.P.'s work was funded by the M.H. Wiener Foundation (ICCP Project). K.N. acknowledges the support provided by the Austrian Science Fund FWF (grant I-1183-N19). The Laboratory of Ion Beam Physics is partially funded by its consortium partners PSI, EAWAG, and EMPA.

## Author contributions

L.W., M.C. and N.B. designed the study with input from A.B., C.L.P., D.B., K.N. and L.W. Radiocarbon measurements and analyses were performed by N.B. and L.W. K.N., T.P., D.B. and A.B. supplied the annually resolved tree-ring samples and are responsible for the documentation of the

dendrochronology. The modeling and interpretation of the [14]C data were done by N.B, M.C., and L.W. with input from C.L.P. N.B. wrote the manuscript with input from all other authors.

## Competing interests

The authors declare no competing interests.
