## [Transparent Peer Review file · Nature Communications]

Tracing Ancient Solar Cycles with Tree Rings and Radiocarbon in the First Millennium BCE

Corresponding Author: Dr Nicolas Brehm

Version 0:

Reviewer comments:

Reviewer #1

(Remarks to the Author)
GENERAL COMMENTS

This is a very interesting study, the number of measurements presented is really impressive, and the fact that they are at a single year resolution is a big contribution to the radiocarbon community. The data analysis looks solid, however I have some questions on that regard and I would like to see a bit more in-depth explanations. Furthermore, some general statements need to be rephrased in my opinion. See the details in my specific comments below.

In general, about citations:

Citations style is inconsistent: sometimes (Line 59, 204, 211, please check all citations in case I missed others) the full paper title is reported together with Authors and Year, while in all other cases not.

SPECIFIC COMMENTS

Line 16

“caused by solar energetic particle (SEP) events”

Maybe just add a “most likely” or similar before this sentence. Despite SEPs are the strongest candidate, there is still no complete consensus on this matter.

Line 32

“Poluiyanov, Kovaltsov, ishev, & Usoskin, 2016”

Misspelling in one of the names, Mishev.

Line 44

“While several annually resolved ^{14}C records have emerged recently, they often fall short in length and analytical precision”

What do you mean by “they fall short in analytical precision”? It sounds like you are implying that those study have unreliable and inconsistent results.

Lines 56-58

“Moreover, it enables the identification and examination of ^{14}C production spikes, which have recently garnered attention due to their association with solar energetic particle (SEP) events”

I don't see how this statement applies to this dataset, since as you mention later, no new production spikes are found in this record.

Line 111 & Lines 122-124 & Lines 127-128

“The 50 yr low-pass filtered reconstruction solar modulation parameter [...]”

“The $\Delta^{14}\text{C}$ data was detrended by subtracting the 30-year lowpass filtered $\Delta^{14}\text{C}$ record. Then a Morlet wavelet transform

was calculated[...]"

"Additionally, a Lomb-Scargle periodogram (VanderPlas, 2018) was computed for different time-ranges of the Δ 14C data."

In all these last three text extracts, I think that more information about these analyses needs to be provided, either before the introduction of those concepts or in an ad-hoc methods section. Why the choice of 50 years as size of the low-pass filter for reconstructing the solar modulation parameter and 30 years for detrending the record? Why applying a Morlet wavelet transform, and why a Lomb-Scargle periodogram? I am not implying that these choices are wrong, but a brief and concise explanation on the reason for these choices would help clarify the analysis process.

Lines 171-175

"The presence of consistent 11-year cycles in the 14C data during times of high solar activity, has an entirely separate 'dual function' in that these data also present major new opportunities to fit short, single year sequences from tree-rings across radiocarbon plateaus. This will significantly improve the precision of dating provided by Bayesian chronological modeling of archaeological or environmental radiocarbon data more widely in this period."

This statement is quite bold, and if you want to include it, I would say that you should show a proof of concept, that is a comparison of calibration of a few dates from the plateau using IntCal20 and your dataset, to prove that this higher resolution actually improves the precision of the calibration.

Lines 211-212

"Typically, a 1‰ external error for sampling and sample preparation was added to the finally reported uncertainties."

Is this value (1‰) an arbitrary choice or does it come from calculated estimates of additional uncertainty introduced by sampling and preparation? Or does it come from literature?

Line 219

Typo, "no repeated measurements were done".

Lines 226-232

"The results given for the 1515 CE reference are very consistent (SI Figure 3) with a mean radiocarbon age of 337.1 ± 1.7 yr BP (F14C: 0.9589 ± 0.0002). The sample scatter of 13 yr is slightly less than the mean uncertainty of 14 yr given by the applied data reduction ($\chi^2 = 61.0$, $n = 69$, $p=0.74$). For the 492 BCE reference a mean radiocarbon age of 2439.7 ± 2.2 yr BP (F14C: 7381 ± 0.0002) was obtained. The sample scatter of 13 yr is also slightly less than the mean uncertainty of 16 yr given by the applied data reduction ($\chi^2 = 44.6$, $n = 52$, $p=0.76$)."

I understand that those mean uncertainties (1.7 and 2.2 years) come from averaging every measurement of each control sample, that have an average uncertainty of 14 and 16 years respectively. Maybe specify this in a more explicit way, as I think this explanation can be a bit confusing. However, even if those ages and relative uncertainties are the outcome of averaging, I think that you should round both age and uncertainty to the closest integer (i.e. 337.1 ± 1.7 becomes 337 ± 2); otherwise you would be overestimating the precision of your measurements, since we are dealing with single tree rings which have an annual resolution as the lower limit of precision.

Lines 248-249

"The initial 12C masses of the global reservoirs taken from Güttler (Güttler, et al., 2015) were distributed accordingly (Supplementary figure boxmodel)."

This "Supplementary figure boxmodel" is not present, neither in the Supplementary Information, nor in the main manuscript. Please provide that!

Lines 327 – 329

"Since the energy spectrum of the SEPs is softer than the one of the galactic cosmic rays the excess 14C produced by the event introduced by the Gaussian spike was mainly put into the stratosphere (90%) and only 10% into the troposphere."

Is this specific choice of 90% and 10% supported by literature or by some calculations, or is that arbitrary?

Reviewer #2

(Remarks to the Author)

Review

Brehm et al. : Tracing the Ancient Sun: Annual Tree Ring Radiocarbon Reveals the 11-Year Solar Cycle in the First Millennium BCE
Nature Communications, Nov. 2024

This is an interesting manuscript bringing new radiocarbon data set and evaluations for annual 14C data in tree rings in the first millennium BCE, which after minor improvements could be published in Nature Communications.

A few suggestions:

Abstract:

20-25:

Include there more quantitative results, e.g. the amplitude of the measured 11-year solar cycle, the main period, etc.

Introduction:

32:

“Poluianov, Kovaltsov, ishev, & Usoskin, 2016”

It would not be enough to report as Poliuanov et al., 2016 (to save a space)?

...and similarly in other citations...

40:

A reference to the first paper (e.g Castagnoli, Lal in Nobel Symp.) in which this approach was introduced should be included.

44:

“While several annually resolved C-14 records have emerged recently, they often fall short in length and analytical precision...”

Probably you mean during the first millennium BCE, otherwise much older citations should be included (starting from seventies, when, generally, people did not believe in 11-year 14C variations...

Results:

133:

...reduced during solar minima.”

A reference is missing!

Discussion:

Actually, this part is very short, and just repeating results presented in the previous section.

I would expect a discussion, e.g., on carbon cycle box modelling, on the amplitudes of the 11-year cycle, etc. (as presented in the Methods section).

Fig. 1b:

“offset 14C age”

I propose to include there “offset 14C age in years”.

Error bars are at 2sigma (it should be also mentioned)?

Fig. 2a:

“normalized 14C production in ??

“solar modulation parameter Φ (MeV)”

Fig. 3e:

“Lomb-Scargle periodogram of the only the time periods of the two grand solar minima.”

I propose to include there: “when the 11-year solar cycles were absent.”

Fig. 4:

In physics the 5sigma threshold is required for the discovery of an event, why did you choose 3sigma?

Supplementary

Presented 14C data set is of high scientific value.

Pavel Povinec

Reviewer #3

(Remarks to the Author)

This paper presents the newly obtained annual carbon-14 record across the first millennium BCE, which serves as a dataset for retrieving past solar cycle information and as a foundation for precise dating. There are a few minor issues as listed below, however, once these are addressed, I recommend publication in this journal.

Comments and suggestions:

Lines 29, 32 and 58 (and many other places):

Please let the references in chronological order.

Line 36-38:

This sentence needs to be modified. The relative flux for positive and negative polarities depends on the level of solar activity which lead to the change in the variability of tilt angle (please see Figure 4 of Kota and Jokipii (1983)). During periods of low solar activity, such as the grand solar minima, the flux can be higher at negative polarities (see Kataoka et al., Space Weather, 2012).

Line 40-41 & Lines 304-305:

Please include the reference "Masarik and Beer, 1999". Also, please put all the references within a single set of parentheses.

Line 43-49:

These introductory sentences need to be comprehensive and precise.

- Line 43-44:

Among the relatively new series, there are a few high precision series such as Miyahara et al. (2021) and Miyahara et al. (2022), which are not cited here. The precisions of those series are in fact better than the data obtained in this study. So, the motivation that should be emphasized here seems to be not the precision, but the length.

- It is better to mention Fahrni et al. (2020) in these introductory sentences as well.

- Line 47-48: "Although these studies already consist of"

This sentence is not clear as the references in the previous sentence do not include Brehm et al. (2021). What about modifying as "Recently, annual 14C tree ring data were obtained over the last millennium, however, no millennia-long annually resolved records exist beyond the first millennium CE".

Line 50:

over 1000 years -> over the first millennium BCE

Line 53:

Figure 1 has been cited here, but it is not clear which data are by Fahrni et al. (2020) in Figure 1. It is better to indicate which are their data both in Figure 1 and Figure 2.

Line 59:

There is no need to include the title of the paper.

Line 63-64: "This reveals new insights into..."

-> These data can reveal new insights into solar cycles and can also be used to...

Line 65: "radiocarbon dating in this time period"

-> radiocarbon dating in this time period, particularly between ca. 750 and 420 BCE, by adding sub-decadal structure to data available for radiocarbon calibration.

Line 88:

Please include the absolute period range measured in this study. There seems some overlap with Fahrni et al. (2020). Are they weighted-averaged?

Figure 1:

Please include in the caption where to find the details of the samples. I realize there is no information regarding "Baunach 33". Please check.

Line 101:

-> carbon cycle 22-box model (Güttler, et al., 2015)

Lines 104-105 & Figure 2:

It is not clear if the error of Delta 14C for the 664 BCE event (or the error for the production rate for this SEP event) is propagated to the data for 663 BCE to 450 BCE. Also, please include the production rate used for the calculation.

Lines 137-138: "This finding was also visible in the wavelet analysis"

It seems that, whereas the Morlet Wavelet analysis reveals significant 22-yr cycle during the grand solar minima (Figure 3(b)), there are no such peak in the Lomb-Scargle power in Figure 3(e). It is therefore not appropriate to say that "This finding was also visible" here. Please note that this discrepancy may be due to the fact that the Lomb-Scargle analysis is not suitable for time series with quasi-periodicity or for cases where the periods have unstable phases (FFT has the same problem). For example, it is possible that the signal of the 11-yr cycle vanishes in the case the phases of the 11-yr cycle are 180 degrees inverted between the two grand minima. This effect can be particularly serious when the data spans are relatively short. I therefore suggest either to exclude the FFT and Lomb-Scargle spectra or to include the caution in the text regarding the limitations of these two methods, and to conduct discussions mainly based on the Morlet Wavelet spectrum.

Line 162-163: "nonexistent"

Please convey the discussions based on the Morlet Wavelet analysis as suggested above. The spectrum certainly suggests the reduction of the cycle amplitude during the grand solar minima, but the existence of 22-year cycle rules out the possibility

of nonexistence of the solar cycles. Please refer to Kataoka et al. (2012) for the significant 22-year cosmic-ray cycles during grand solar minima, that can be caused by continued (but reduced) solar cycles. Please also note that previous works have consistently suggested the continuity of solar cycles during the grand minima (Beer et al., 1998; Usoskin and Mursula, 2001; Miyahara et al., 2004; Brehm et al., 2021).

Table 1:

Of the 11 obtained p-values -> Of the 11 obtained p-values,
don -> done

Lines 234-299:

It was unclear whether the model is identical to that of Brehm et al. (2021). If there have been any updates, please summarize them. If not, please cite Brehm et al. (2021) in the first or second paragraph. Additionally, please specify which set of the carbon exchange rates (Fij and Fji) was used in this study (are they as the same as in Extended Data Fig. 3 of Brehm et al. (2021)?).

Line 236:

$\Delta^{14}\text{C}$ -> ^{14}C

Line 239:

For this -> For this,

Line 249: "Supplementary figure boxmodel"

Are you referring Appendix A in Güttler et al. (2015)?

Line 312:

Please specify which table.

Figure 3(a):

Please indicate the color bar for the contour.

Version 1:

Reviewer comments:

Reviewer #1

(Remarks to the Author)

Dear authors, thank you for taking my previous comments into account and for addressing them thoroughly. After the latest revision of the manuscript, my main questions and concerns have been answered and resolved properly, therefore I have no additional comments to provide, and I recommend the manuscript to be published.

Best regards

Reviewer #2

(Remarks to the Author)

The paper has been improved and it can be accepted for publication.

Detailed reply to the reviewer's comments

We sincerely thank all the reviewers for their thoughtful and constructive comments. Your valuable feedback has been instrumental in refining and improving the quality of our work. We deeply appreciate the time and effort you dedicated to reviewing our manuscript.

Here we give a detailed response to all the issues and comments raised by the reviewers. The reviewer's comments are displayed in black while our replies are written in blue.

Reviewer #1 (Remarks to the Author):

GENERAL COMMENTS

This is a very interesting study, the number of measurements presented is really impressive, and the fact that they are at a single year resolution is a big contribution to the radiocarbon community. The data analysis looks solid, however I have some questions on that regard and I would like to see a bit more in-depth explanations. Furthermore, some general statements need to be rephrased in my opinion. See the details in my specific comments below.

In general, about citations:

Citations style is inconsistent: sometimes (Line 59, 204, 211, please check all citations in case I missed others) the full paper title is reported together with Authors and Year, while in all other cases not.

We have now changes the citation style to numbered citations to resolve this issue.

SPECIFIC COMMENTS

Line 16

“caused by solar energetic particle (SEP) events”

Maybe just add a “most likely” or similar before this sentence. Despite SEPs are the strongest candidate, there is still no complete consensus on this matter.

Thanks for the suggestion! We implemented it.

Line 32

“Poluianov, Kovaltsov, ishev, & Usoskin, 2016”

Misspelling in one of the names, Mishev.

Was corrected

Line 44

“While several annually resolved 14C records have emerged recently, they often fall short in length and analytical precision”

What do you mean by “they fall short in analytical precision”? It sounds like you are implying that those study have unreliable and inconsistent results.

You are right! The sentence really implies low quality data, which is not true. We now just write: “While several annually resolved ^{14}C records have emerged recently, they often fall short in length, ...”

Lines 56-58

“Moreover, it enables the identification and examination of ^{14}C production spikes, which have recently garnered attention due to their association with solar energetic particle (SEP) events”

I don't see how this statement applies to this dataset, since as you mention later, no new production spikes are found in this record.

We agree, thank you and have changed this as follows: “Moreover, it allows us to systematically test for the possibility of new ^{14}C production spikes (which have recently garnered attention due to their association with solar energetic particle (SEP) events) within this period...”

Line 111 & Lines 122-124 & Lines 127-128

“The 50 yr low-pass filtered reconstruction solar modulation parameter [...]”

“The $\Delta^{14}\text{C}$ data was detrended by subtracting the 30-year lowpass filtered $\Delta^{14}\text{C}$ record. Then a Morlet wavelet transform was calculated[...]

“Additionally, a Lomb-Scargle periodogram (VanderPlas, 2018) was computed for different time-ranges of the $\Delta^{14}\text{C}$ data.”

In all these last three text extracts, I think that more information about these analyses needs to be provided, either before the introduction of those concepts or in an ad-hoc methods section. Why the choice of 50 years as size of the low-pass filter for reconstructing the solar modulation parameter and 30 years for detrending the record? Why applying a Morlet wavelet transform, and why a Lomb-Scargle periodogram? I am not implying that these choices are wrong, but a brief and concise explanation on the reason for these choices would help clarify the analysis process.

We now added short and concise reasons for the choices of analysis:

“For comparison with other lower resolution records of solar modulation parameters (Steinhilber, et al., 2012; Wu, et al., 2018), we used a 50 year lowpass filter for our reconstruction to match the resolution of these records.”

“To limit the analysis to the short periods of interest, such as the 11-year and 22- year cycles, the $\Delta^{14}\text{C}$ data was detrended by subtracting ...”

“To get information about the amplitude and frequency of the 11-year solar cycle through time a Morlet wavelet transform was calculated, where the areas of more than 95 % significance were marked.”

“To investigate this further, a Lomb-Scargle analysis, well known for the detection and characterization of periodic signals in unevenly sampled data (VanderPlas, 2018), was applied solely on the data from 833 BCE to 705 BCE and 413 to 325 BCE, covering the two grand solar minima within the study period.”

Lines 171-175

“The presence of consistent 11-year cycles in the ^{14}C data during times of high solar activity, has an entirely separate ‘dual function’ in that these data also present major new opportunities

to fit short, single year sequences from tree-rings across radiocarbon plateaus. This will significantly improve the precision of dating provided by Bayesian chronological modeling of archaeological or environmental radiocarbon data more widely in this period.”

This statement is quite bold, and if you want to include it, I would say that you should show a proof of concept, that is a comparison of calibration of a few dates from the plateau using IntCal20 and your dataset, to prove that this higher resolution actually improves the precision of the calibration.

We now changed the statement to: “The presence of consistent 11-year cycles in the ¹⁴C data during times of high solar activity has an entirely separate ‘dual function’ in that these data also present major new opportunities to fit short, single year sequences from tree-rings across radiocarbon plateaus. This could significantly improve the precision of dating provided by Bayesian chronological modeling of archaeological or environmental studies “

Lines 211-212

“Typically, a 1‰ external error for sampling and sample preparation was added to the finally reported uncertainties.”

Is this value (1‰) an arbitrary choice or does it come from calculated estimates of additional uncertainty introduced by sampling and preparation? Or does it come from literature?

You are right, we did not explain why the 1‰ external error was added. We now additionally write: “The additional 1‰ was estimated by the long-term evaluation of the references used for quality control.”

Line 219

Typo, “no repeated measurements were done”.

Was corrected.

Lines 226-232

“The results given for the 1515 CE reference are very consistent (SI Figure 3) with a mean radiocarbon age of 337.1 ± 1.7 yr BP (F14C: 0.9589 ± 0.0002). The sample scatter of 13 yr is slightly less than the mean uncertainty of 14 yr given by the applied data reduction ($\chi^2 = 61.0$, $n = 69$, $p=0.74$). For the 492 BCE reference a mean radiocarbon age of 2439.7 ± 2.2 yr BP (F14C: 7381 ± 0.0002) was obtained. The sample scatter of 13 yr is also slightly less than the mean uncertainty of 16 yr given by the applied data reduction ($\chi^2 = 44.6$, $n = 52$, $p=0.76$).”

I understand that those mean uncertainties (1.7 and 2.2 years) come from averaging every measurement of each control sample, that have an average uncertainty of 14 and 16 years respectively. Maybe specify this in a more explicit way, as I think this explanation can be a bit confusing. However, even if those ages and relative uncertainties are the outcome of averaging, I think that you should round both age and uncertainty to the closest integer (i.e. 337.1 ± 1.7 becomes 337 ± 2); otherwise you would be overestimating the precision of your measurements, since we are dealing with single tree rings which have an annual resolution as the lower limit of precision.

We apologize for the confusion. We now explicitly say how the uncertainties have been calculated by writing:

“, where the uncertainty of the mean was estimated by calculating the error of the mean.”

We additionally also changed the precision of the reported values as suggested, to not overestimate the precision of the measurements.

Lines 248-249

“The initial ^{12}C masses of the global reservoirs taken from Güttler (Güttler, et al., 2015) were distributed accordingly (Supplementary figure boxmodel).”

This “Supplementary figure boxmodel” is not present, neither in the Supplementary Information, nor in the main manuscript. Please provide that!

We apologize for forgetting to add the figure to the supplements. We now Supplied the depiction of the used radiocarbon box model to the supplements (Figure 4).

Lines 327 – 329

“Since the energy spectrum of the SEPs is softer than the one of the galactic cosmic rays the excess ^{14}C produced by the event introduced by the Gaussian spike was mainly put into the stratosphere (90%) and only 10% into the troposphere.”

Is this specific choice of 90% and 10% supported by literature or by some calculations, or is that arbitrary?

Sorry for not clarifying this choice. We now explain this by writing:

“The choice of 90% stratospheric and 10% tropospheric production were made by rounding the calculated values from ^{14}C production simulations (12.05 % Troposphere and 87.95%) (Golubenko, Rozanov, Kovaltsov, & Usoskin, 2022) with the assumption of a “hard spectrum” as proposed for the 664 BCE event by O’Hare et al. (O’Hare, et al., 2019).“

Reviewer #2 (Remarks to the Author):

Review

Brehm et al. : Tracing the Ancient Sun: Annual Tree Ring Radiocarbon Reveals the 11-Year Solar Cycle in the First Millennium BCE
Nature Communications, Nov. 2024

This is an interesting manuscript bringing new radiocarbon data set and evaluations for annual ^{14}C data in tree rings in the first millennium BCE, which after minor improvements could be published in Nature Communications.

A few suggestions:

Abstract:

20-25:

Include there more quantitative results, e.g. the amplitude of the measured 11-year solar cycle, the main period, etc.

Thanks for the suggestion. We now included a characterization of the cycle length and amplitude and write:

“New SEP events were ruled out and the 11-year cycle was revealed during times of high solar activity with a mean period of 10.5 years. The full record reveals a mean amplitude of about 0.4‰ in ¹⁴C concentration due to the 11-year solar cycle.” In the abstract and also added: “A simple fast Fourier transform on the detrended data also reveals a significant (> 99.9 % false alarm level) mean period of about 10.5 years with an amplitude of about 0.4‰ (Figure 3c).”

Introduction:

32:

“Poluianov, Kovaltsov, ishev, & Usoskin, 2016”

It would not be enough to report as Poliuanov et al., 2016 (to save a space)?

...and similarly in other citations...

We chose to have the citations in plain text for a better overview for the review process. We now changed the citation style to just numbers like the style of Nature Communication is.

40:

A reference to the first paper (e.g Castagnoli, Lal in Nobel Symp.) in which this approach was introduced should be included.

Thank you for the suggestion. We now added (Lal & Peters, 1967) and (Masarik & Beer, 1999) as a reference.

44:

“While several annually resolved C-14 records have emerged recently, they often fall short in length and analytical precision...”

Probably you mean during the first millennium BCE, otherwise much older citations should be included (starting from seventies, when, generally, people did not believe in 11-year ¹⁴C variations...

We apologies for the confusion. We now changed this section and now write:

“While several annually resolved ¹⁴C records have emerged recently, these have been limited in length (Menjo, et al., 2005; Miyahara, et al., 2007; Miyahara, et al., 2008; Hong, et al., 2013; Eastoe, Tucek, & Touchan, 2019; Friedrich, et al., 2019; Moriya, et al., 2019; 2021; Miyahara, et al., 2021,2022; Sakamoto, Hakozaiki, Nakatsuka, & Ozaki, 2023), hindering systematic studies of short-term solar variability such as 11-year and 22-year cycles in the past. Recently, consecutive year annual ¹⁴C tree ring data were obtained over the last millennium (Brehm, et al., 2021), however, so far no millennia-long annually resolved records are available beyond the first millennium CE.”

Results:

133:

...reduced during solar minima.”

A reference is missing!

We modified the text accordingly: “It has been observed that the 11-year solar cycle in ^{14}C tree ring data is significantly reduced during solar minima (Brehm, et al., 2021).”

Discussion:

Actually, this part is very short, and just repeating results presented in the previous section. I would expect a discussion, e.g., on carbon cycle box modelling, on the amplitudes of the 11-year cycle, etc. (as presented in the Methods section).

We now extended the discussion a little bit.

Fig. 1b:

“offset ^{14}C age”

I propose to include there “offset ^{14}C age in years”.

Error bars are at 2sigma (it should be also mentioned)?

We changed the label to “offset ^{14}C age in years” and also clarified the error bars in the caption

Fig. 2a:

“normalized ^{14}C production in ??

“solar modulation parameter Φ (MeV)”

We now mention in the caption that the ^{14}C production is normalized to the steady state production rate of 6.6kg /year. And we also adjusted the labels like you suggested.

Fig. 3e:

“Lomb-Scargle periodogram of the only the time periods of the two grand solar minima.”

I propose to include there: “when the 11-year solar cycles were absent.”

We now write . “e) Lomb-Scargle periodogram of the time periods of the two grand solar minima.”

Fig. 4:

In physics the 5sigma threshold is required for the discovery of an event, why did you choose 3sigma?

You are right that the 3-sigma threshold is kind of an arbitrary choice and technically not a perfect choice. This is not a level of significance that unanimously identifies an event, it’s a threshold. It has to be chosen high enough to exclude events triggered by noise. But it is also mentionable that the 664 BCE event crosses the 3-sigma threshold for two consecutive years. We now rephrased this to: “The most significant ^{14}C increase was detected in the year 663 BCE, which corresponds to the event already found in 664 BCE. This increase passes the 3- σ threshold of 4.5‰ for two consecutive years.”

Supplementary

Presented ^{14}C data set is of high scientific value.

Pavel Povinec

Reviewer #3 (Remarks to the Author):

This paper presents the newly obtained annual carbon-14 record across the first millennium BCE, which serves as a dataset for retrieving past solar cycle information and as a foundation for precise dating. There are a few minor issues as listed below, however, once these are addressed, I recommend publication in this journal.

Comments and suggestions:

Lines 29, 32 and 58 (and many other places):

Please let the references in chronological order.

Thanks for the suggestion. We now put all the references in chronological order.

Line 36-38:

This sentence needs to be modified. The relative flux for positive and negative polarities depends on the level of solar activity which lead to the change in the variability of tilt angle (please see Figure 4 of Kota and Jokipii (1983)). During periods of low solar activity, such as the grand solar minima, the flux can be higher at negative polarities (see Kataoka et al., Space Weather, 2012).

Because this discussion seemed a bit too complicated for the first few introductory sentences we decided to shift the discussion of why the 22-year cycles are visible in our data to the discussion section of the paper. We now write in the discussion: "This confirms the model, that the relative flux for positive and negative polarities depends on the level of solar activity which leads to the change in the variability of tilt angle (Kota & Jokipii, 1983). During periods of low solar activity, such as the grand solar minima, the flux can be even higher at negative polarities (Kataoka, Miyahara, & Steinhilber, 2012). This could explain why we can still detect a 22-year cycle during the solar minima when no significant 11-year cycles are detected."

Line 40-41 & Lines 304-305:

Please include the reference "Masarik and Beer, 1999". Also, please put all the references within a single set of parentheses.

Thanks for pointing that out. We now added the reference and corrected the formatting error.

Line 43-49:

These introductory sentences need to be comprehensive and precise.

We made some changes to the introduction also due to some other reviewer comments and made it more concise.

- Line 43-44:

Among the relatively new series, there are a few high precision series such as Miyahara et al. (2021) and Miyahara et al. (2022), which are not cited here. The precisions of those series are in

fact better than the data obtained in this study. So, the motivation that should be emphasized here seems to be not the precision, but the length.

We now added the citations for the high precision records of Miyahara et al 2021 and 2022. Additionally, we now also only mention the lack of long ¹⁴C records and just write:

“While several annually resolved 14C records have emerged recently, they often fall short in length (Menjo, et al., 2005; Miyahara, et al., 2007; Miyahara, et al., 2008; Hong, et al., 2013; Eastoe, Tucek, & Touchan, 2019; Friedrich, et al., 2019; Moriya, et al., 2019; Miyahara, et al., Gradual onset of the Maunder Minimum revealed by high-precision carbon-14 analyses, 2021; Miyahara, et al., Recurrent Large-Scale Solar Proton Events Before the Onset of the Wolf Grand Solar Minimum, 2022; Sakamoto, Hakozaiki, Nakatsuka, & Ozaki, 2023), hindering systematic studies of short-term solar variability such as 11-year and 22-year cycles in the past.

- It is better to mention Fahrni et al. (2020) in these introductory sentences as well.

We now added Fahrni et al. (2020) to the introductory sentence.

- Line 47-48: “Although these studies already consist of”

This sentence is not clear as the references in the previous sentence do not include Brehm et al. (2021). What about modifying as “Recently, annual 14C tree ring data were obtained over the last millennium, however, no millennia-long annually resolved records exist beyond the first millennium CE”.

Thanks for the suggestion. We now changed the sentence to your suggested sentence, which actually expresses what we wanted to say more clearly.

Line 50:

over 1000 years -> over the first millennium BCE

Was implemented

Line 53:

Figure 1 has been cited here, but it is not clear which data are by Fahrni et al. (2020) in Figure 1. It is better to indicate which are their data both in Figure 1 and Figure 2.

We now labeled the data from Fahrni et al (2020) separately in Figure 1

Line 59:

There is no need to include the title of the paper.

We have now changed the citation style to numerical, which is also the style that Nature communication uses.

Line 63-64: “This reveals new insights into...”

-> These data can reveal new insights into solar cycles and can also be used to...

We implemented the phrase how suggested.

Line 65: “radiocarbon dating in this time period”

-> radiocarbon dating in this time period, particularly between ca. 750 and 420 BCE, by adding sub-decadal structure to data available for radiocarbon calibration.

We implemented the phrase how suggested.

Line 88:

Please include the absolute period range measured in this study. There seems some overlap with Fahrni et al. (2020). Are they weighted-averaged?

We now clarify the time periods measured in this study explicitly and write:

“The data cover the two time periods from 1000 BCE to 851 BCE and 652 BCE to 2 BCE at annual resolution and were combined with a previously published single year record covering the period 856 to 626 BCE (17) to produce a 1000-yr-long sequence (1000 BCE – 2 BCE) (Figure 1).”

Figure 1:

Please include in the caption where to find the details of the samples. I realize there is no information regarding “Baunach 33”. Please check.

The Baunach 33, Oberhaid 15 and Trieb 70 a data was published in Fahrni et al. (2020). We now clarify this by explicitly labeling this data with Fahrni et al. (2020) in Figure 1.

Line 101:

-> carbon cycle 22-box model (Güttler, et al., 2015)

Thanks for the suggestion. Since the model used is the same as Brehm et al. 2021 we referenced that. Güttlers model only used 11 boxes.

Lines 104-105 & Figure 2:

It is not clear if the error of Delta 14C for the 664 BCE event (or the error for the production rate for this SEP event) is propagated to the data for 663 BCE to 450 BCE. Also, please include the production rate used for the calculation.

We did not propagate any additional error to the data for the 664 BCE event. We now write that no error was propagated to the corrected data by writing: “No additional error was propagated to the detrended data.”

Lines 137-138: “This finding was also visible in the wavelet analysis”

It seems that, whereas the Morlet Wavelet analysis reveals significant 22-yr cycle during the grand solar minima (Figure 3(b)), there are no such peak in the Lomb-Scargle power in Figure 3(e). It is therefore not appropriate to say that “This finding was also visible” here. Please note that this discrepancy may be due to the fact that the Lomb-Scargle analysis is not suitable for time series with quasi-periodicity or for cases where the periods have unstable phases (FFT has the same problem). For example, it is possible that the signal of the 11-yr cycle vanishes in the case the phases of the 11-yr cycle are 180 degrees inverted between the two grand minima. This effect can be particularly serious when the data spans are relatively short. I therefore suggest either to exclude the FFT and Lomb-Scargle spectra or to include the caution in the text regarding the limitations of these two methods, and to conduct discussions mainly based on the Morlet Wavelet spectrum.

We do not completely agree with the reviewer here. Although it is true that the Lomb-scargle analysis is not perfectly suitable for this kind of analysis, the result could not be reproduced by

selecting random sections with the same length as the two solar minima. To clarify this, we now also included a short section about the limitations of the analysis:

“While these observations seem compelling, we note that Lomb-Scargle analysis has some limitations for the evaluation of quasi-periodic signals such as the 11—year cycle. We also note that while the appearance of 22-year cycles in the wavelet analysis seems to potentially contradict an argument for the absence of 11-year cycles, instead it may indicate that one of the solar magnetic polarities simply has a significantly stronger shielding effect from the galactic cosmic rays than the other.”

Line 162-163: “nonexistent”

Please convey the discussions based on the Morlet Wavelet analysis as suggested above. The spectrum certainly suggests the reduction of the cycle amplitude during the grand solar minima, but the existence of 22-year cycle rules out the possibility of nonexistence of the solar cycles. Please refer to Kataoka et al. (2012) for the significant 22-year cosmic-ray cycles during grand solar minima, that can be caused by continued (but reduced) solar cycles. Please also note that previous works have consistently suggested the continuity of solar cycles during the grand minima (Beer et al., 1998; Usoskin and Mursula, 2001; Miyahara et al., 2004; Brehm et al., 2021).

See comment above

Table 1:

Of the 11 obtained p-values -> Of the 11 obtained p-values,
don -> done

Was corrected

Lines 234-299:

It was unclear whether the model is identical to that of Brehm et al. (2021). If there have been any updates, please summarize them. If not, please cite Brehm et al. (2021) in the first or second paragraph. Additionally, please specify which set of the carbon exchange rates (F_{ij} and F_{ji}) was used in this study (are they as the same as in Extended Data Fig. 3 of Brehm et al. (2021)?).

The model is indeed the same as in Brehm et al. (2021). We now explicitly refer to Brehm et al. (2021). And now also supplied the supplementary figure with all the fluxes and carbon contents of each box.

Line 236:

$\Delta^{14}\text{C}$ -> ^{14}C

Was corrected.

Line 239:

For this -> For this,

Was corrected.

Line 249: “Supplementary figure boxmodel”

Are you referring Appendix A in Gütthler et al. (2015)?

We just forgot to add the figure to the supplements. The figure is now supplied in the supplements and correctly referenced

Line 312:

Please specify which table.

We now clarified the statement and show the relationship between production rate, solar modulation and VADM in a figure in the supplements. We now write:

“For a given VADM and ^{14}C production rate, the solar modulation parameter Φ can be estimated by considering the relation of production rate, solar modulation parameter and VADM that was computed by Herbst et al. (Herbst, Muscheler, & Heber, 2017)(Supplementary Figure 5).”

Figure 3(a):

Please indicate the color bar for the contour.

We now added the color bar for the contour.

Detailed reply to the reviewer's comments

Here we give a detailed response to all the issues and comments raised by the reviewers. The reviewer's comments are displayed in black while our replies are written in blue.

Reviewer #1 (Remarks to the Author):

Dear authors, thank you for taking my previous comments into account and for addressing them thoroughly. After the latest revision of the manuscript, my main questions and concerns have been answered and resolved properly, therefore I have no additional comments to provide, and I recommend the manuscript to be published.

Best regards

Reviewer #2 (Remarks to the Author):

The paper has been improved and it can be accepted for publication.

Dear Reviewers,

Thank you for your positive feedback and for your recommendation to publish our manuscript. We greatly appreciate your time, effort, and constructive suggestions throughout the review process, which have significantly improved the quality of our work.

Best regards,
Nicolas Brehm